# Text Embeddings Should Capture Implicit Semantics, Not Just Surface Meaning

## Abstract

*This position paper argues that the text embedding research community should move beyond surface meaning and embrace implicit semantics as a central modeling goal.* Text embedding models have become foundational in modern NLP, powering a wide range of applications and drawing increasing research attention. Yet, much of this progress remains narrowly focused on surface-level semantics. In contrast, linguistic theory emphasizes that meaning is often implicit, shaped by pragmatics, speaker intent, and sociocultural context. Current embedding models are typically trained on data that lacks such depth and evaluated on benchmarks that reward the capture of surface meaning. As a result, they struggle with tasks requiring interpretive reasoning, speaker stance, or social meaning. Our pilot study highlights this gap, showing that even state-of-the-art models perform only marginally better than simplistic baselines on implicit semantics tasks. To address this, we call for a paradigm shift: embedding research should prioritize more diverse and linguistically grounded training data, design benchmarks that evaluate deeper semantic understanding, and explicitly frame implicit meaning as a core modeling objective, better aligning embeddings with real-world language complexity.

## 1 Introduction

Text embedding models are designed to transform textual content, whether sentences, paragraphs, or full documents, into dense vectors in a high-dimensional space, where the proximity of embeddings reflects semantic similarity [123, 102]. These models have become foundational in modern NLP and are now widely deployed in a pre-trained, off-the-shelf manner across a wide range of downstream tasks such as clustering [41, 8], classification [102], information retrieval [148, 57], and retrieval-augmented generation (RAG) [76]. In response, the research community has dedicated extensive effort to improving model architectures [123, 86, 12, 101], training strategies [38, 150, 82, 163], and evaluation benchmarks [102, 44, 34].

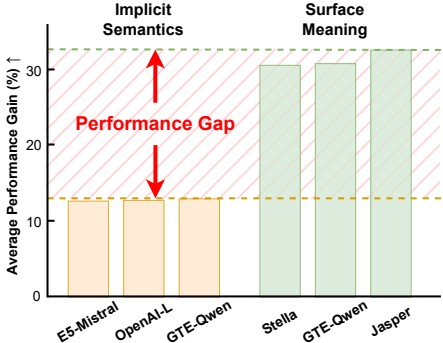

Figure 1: Average performance gains of top embedding models over the Bag-of-Tokens baseline on two evaluation sets: implicit semantics (averaged over seven datasets from Table 1) and surface meaning (averaged over MTEB classification tasks [102]).

**The Overlooked Implicit Semantics** Despite substantial advances in text embedding research, a critical gap remains: most embedding models are designed to capture surface-level semantics, such

as lexical overlap, syntactic variation, and topical similarity, while largely neglecting the deeper, implicit layers of meaning that are fundamental to human communication. Decades of linguistic theory have shown that meaning is often shaped not just by what is explicitly stated, but by what is implied, presupposed, or embedded within cultural and social context [50, 91, 63, 131, 17]. These Implicit meanings (e.g., pragmatic intent, speaker stance, and ideological framing) play a crucial role in how language is interpreted, shaping meaning in ways that go far beyond surface form.

**Why Current Models Miss Implicit Meaning** Yet, current embedding models are not designed to capture these rich and nuanced aspects of meaning. This limitation stems from two core issues: training data rarely provides supervision for implicit meaning, and benchmarks do not evaluate or reward its capture. Most embedding models are trained on datasets optimized for surface-level similarity, particularly those derived from information retrieval tasks [9, 70], which offer little opportunity to learn context-sensitive or socially grounded semantics. Compounding this issue, widely adopted benchmarks rarely test for deeper interpretive capabilities [148, 102], further disincentivizing the development of models that aim to go beyond shallow semantic matching. As a result, even state-of-the-art embeddings often fall short in capturing the implicit dimensions of language that are essential for human-like understanding.

**The Performance Divide** To investigate this limitation, we conduct a pilot study using a suite of linguistically informed datasets covering three tiers of implicit meaning: (1) utterance level (pragmatic inference), (2) speaker level (stance), and (3) society level (political and social bias). The empirical results reveal that state-of-the-art embedding models, despite excelling on conventional benchmarks, perform only marginally better than the Bag-of-Tokens baseline on tasks requiring implicit understanding. As illustrated in Figure 1, there is a substantial performance gap between models' capabilities to capture surface meaning versus implicit semantics.

**Our Position** **We argue that the text embedding research community must move beyond surface-level semantics and explicitly embrace implicit meaning as a core modeling objective.** This position paper calls for a shift in research priorities–toward curating more linguistically grounded training data, developing benchmarks that evaluate deeper semantic and social understanding, and building embedding models that more faithfully reflect the complexity of human communication.

# 2 Linguistic Foundations of Implicit Meaning

To sharpen our understanding, we first revisit the linguistic foundations of implicit meaning through a three-tier framework: utterance (pragmatics), speaker (stance-taking), and society (sociolinguistics).

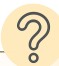
**Research Question: Implicit Semantics**

How do linguistic theories shape implicit meaning?

## 2.1 Utterance Level: Linguistic Signals of Implicit Meaning

Pragmatics investigates how utterances derive meaning from context, bridging the gap between literal surface semantics and the speaker's intended message [40, 50, 91]. It foregrounds what is left unsaid yet successfully communicated, revealing interpretive layers that semantic analysis alone cannot fully capture. This perspective has significantly influenced NLP, particularly in tasks requiring deeper contextual reasoning [47, 18].

At the heart of pragmatics is the insight that meaning emerges from broader situational, social, and cultural contexts, including shared background knowledge and prevailing norms, which collectively guide interpretation [50, 91]. Within this framework, speakers frequently rely on **implicature**, indirect cues inferred rather than explicitly stated [40, 119, 48, 91]. For instance, the sentence "*Bart managed to pass the test*" subtly suggests his success was unexpected, though not logically entailed.

Another key construct is **presupposition**, where utterances embed background assumptions required for comprehension [119, 91]. A statement like "*Sam quit smoking*" presupposes that Sam smoked before, an assumption that persists under negation or interrogation. Together, these phenomena demonstrate how implicit meaning arises not only from what is said, but also from what is assumed or inferred–posing a foundational challenge for text embeddings aiming to model such nuance.

## 2.2 Speaker Level: Cognitive Processes in Implicit Meaning

While pragmatics focuses on utterances in context, the concept of **stance** emphasizes the speaker's internal positioning–expressing attitudes, evaluations, and degrees of alignment or commitment [63]. Stance-taking is crucial to implicit meaning, as it reveals emotional and social orientation through subtle linguistic cues. Kiesling's model formalizes stance through three dimensions: evaluation (positive or negative appraisal), alignment (social positioning relative to others), and investment (the degree of speaker commitment) [32, 75].

Sociolinguistic variation often reflects stance. Forms like *-in'* vs. *-ing* sever not only as dialectal variants but as markers of toughness, informality, or solidarity [62, 152]. Over time, such forms become enregistered–decoupled from specific groups and reused more broadly to index stance. For example, the word *dude* has shifted from a gendered term to a marker of casual camaraderie [61]. Quantitative studies further reveal that stance fluctuates across discourse, with dynamic shifts in speaker intent and alignment observed in corpora like Reddit [64]. In short, stance introduces a relational, affective, and indexical layer to meaning–complementing pragmatics and posing a challenge for embeddings to capture speaker intent and social positioning.

## 2.3 Society Level: Cultural Shaping of Implicit Meaning

Beyond individual cognition and utterances, **sociolinguistics** explores how meaning is shaped by identity, power, and culture. Variation in pronunciation, grammar, or vocabulary, such as dropping the *g* in *workin'*, regional vowel shifts, or particles like *lah* in Singapore English, serves as a social index, signaling class, peer-group belonging, or regional identity [131]. These features are culturally contingent: the same form may index friendliness in one context and lack of education in another. Embedding models that collapse such variation into surface-level representations risk erasing these nuanced social signals.

Language ideologies further complicate this picture by privileging certain varieties while stigmatizing others [15]. As high-status registers dominate pretraining corpora, embeddings often reflect and amplify social hierarchies. For example, African-American Vernacular English may be marginalized relative to Standard American English, encoding structural inequalities as statistical artifacts. Speakers also fluidly shift styles–alternating registers, dialects, or slang–to perform identity and negotiate relationships [17]. These shifts carry implicit social meaning, signaling inclusion, authority, or deference. Yet static embeddings, which average across usage, struggle to capture the fast-paced recalibration of meaning. To reflect the social dimension of language, embeddings must account for the implicit cultural cues embedded in linguistic variation.

> 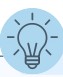 **Takeaway: Linguistic Layers of Implicit Meaning**
>
> Implicit meaning unfolds across three interconnected linguistic tiers: (1) **pragmatics** captures what is implied but unsaid at the utterance level; (2) **stance-taking** reveals the speaker's evaluative, relational, and intentional positioning; and (3) **sociolinguistics** exposes how language encodes identity, culture, and power. Together, these layers illustrate that meaning is not fixed or literal, but deeply contextual, socially embedded, and dynamically performed, posing a significant challenge for embedding models still anchored in surface-level representations.

## 3 Text Embedding Models

Text embedding, the task of mapping text into dense vector representations, has long been central to NLP and now underpins many state-of-the-art applications. This section surveys the evolution of embedding models, highlights active research directions, and critically examines the field's current limitations. Figure 2 provides an overview of the major model classes and trending topics shaping the current embedding landscape.

> 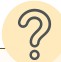 **Research Question: Research Focus**
>
> What is the current state of research on text embedding models?

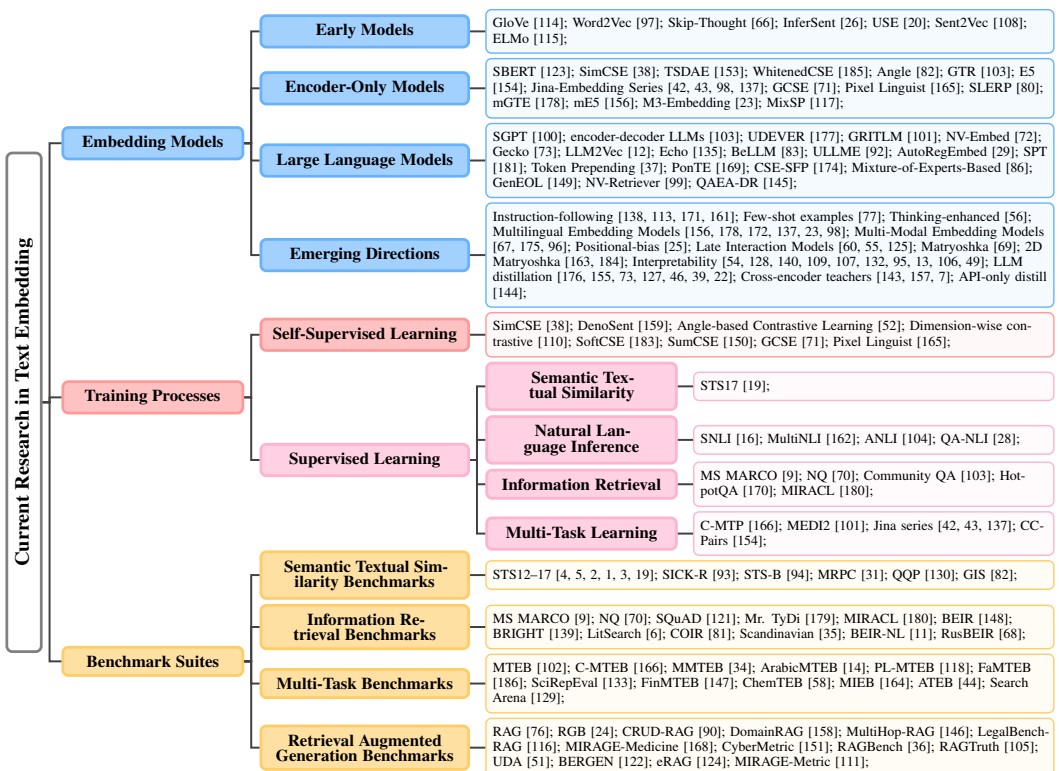

Figure 2: Taxonomy of current research in text embedding.

**Early Models** Initial approaches rely on static word vectors pooled into sentence-level representations, such as Word2Vec and GloVe [97, 114]. Later models like Skip-Thought [66], InferSent [26], Sent2Vec [108], and the Universal Sentence Encoder [20] explore recurrent, transformer, or bilinear architectures. ELMo [115] marks a shift toward contextualized embeddings, dynamically encoding word meaning based on surrounding context.

**Encoder-Only Models** Pretrained encoder-only Transformers like BERT [30] and RoBERTa [88] enable context-aware sentence embeddings via [CLS] or mean pooling, often optimized with contrastive or denoising objectives. Subsequent advances, including Sentence-BERT [123], SimCSE [38], TSDAE [153], and E5 [154], improved embedding quality through new training strategies and architecture refinements [185, 82, 67, 71, 165, 80, 117, 137, 23, 98, 42, 43].

**Large Language Models (LLMs)** LLMs have recently been adapted for embedding tasks using both decoder-only and encoder-decoder designs. Research has explored fine-tuning, prompting, and other strategies to adapt general-purpose LLMs for dense, semantically rich representations. This includes efforts to repurpose decoder-only LLMs [100], adapt encoder-decoder frameworks [103], or build new LLM-based embedding models [101, 72, 73, 12, 135, 83, 92, 29, 181, 37, 169, 174, 86]. While these models offer strong semantic capabilities, their large size and high inference cost limit real-world deployment, sustaining demand for lighter encoder-based alternatives.

**Emerging Directions** Several trends have shaped recent progress. Instruction-following [138, 113, 171, 161], few-shot embedding [77], and "thinking-enhanced" representations [56] aim to improve adaptability. Multilingual and cross-lingual models [156, 178, 172, 137, 23, 98] expand embedding utility across languages. New forms of architectural and training design continue to appear, including sentence embeddings with hypernetworks [171], mitigation of positional bias [25], and efficiency-oriented models like ColBERT [60, 125, 55], Matryoshka [69], and 2D-layered embeddings [163, 184]. Interpretability remains a growing concern, with work exploring human-understandable embeddings [54, 128, 140, 109, 107, 132, 95, 13, 106, 49, 141].

Another emerging direction involves distilling knowledge from LLMs into lightweight sentence embedding models. This includes generating or augmenting training data with LLMs [176, 155, 73, 127, 46, 39], as well as distilling from more accurate but slower cross-encoder models [143, 157, 7].

Some approaches further leverage sentence summaries [150] or distill from proprietary APIs [144]. While these techniques expand supervision and improve performance, they mostly reinforce surface-level semantics, with limited attention to implicit meaning.

**Open Questions: What Should Embeddings Capture?** Despite rapid progress, a central question remains underexplored: what should text embeddings truly capture? While current models excel at encoding surface-level semantics for benchmark-driven tasks, it is less clear whether they can represent more nuanced dimensions such as speaker stance, social context, or pragmatic intent. In the following, we argue that implicit semantics remains a significantly underexplored dimension in the training and evaluation of text embeddings.

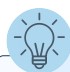 **Takeaway: Landscape of Text Embedding Models**

Research on text embedding models spans architectures, multilinguality, interpretability, and efficiency. Yet, the capacity to capture implicit semantics, which is central to real-world meaning, remains a significantly underexplored frontier.

# 4 Training Processes Fail to Capture Implicit Semantics

Despite significant progress in text embedding models, most training methods remain limited to capture implicit meaning. This section examines the two dominant paradigms: self-supervised and supervised learning, highlighting how both rely on datasets and objectives that prioritize surface-level semantics, leaving deeper contextual and social meanings underrepresented.

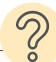 **Research Question: Training Gap**

How are text embedding models trained, and why does this fail to capture implicit meaning?

## 4.1 Self-Supervised Learning

Self-supervised learning trains on unlabeled text by extracting signals through augmentation or structural cues, without requiring manual labels. Techniques like SimCSE [38] uses dropout noise to create sentence pairs, while DenoSent [159] applies a denoising objective. Other approaches explore novel formulations, such as angle-based learning [52], dimension-wise contrastive loss [110], and similarity-weighted negative sampling [183]. Although these methods avoid costly annotations, they generally underperform compared to supervised approaches. Consequently, many embedding models adopt a two-stage pipeline: self-supervised pre-training followed by supervised fine-tuning [38].

## 4.2 Supervised Learning

Supervised training typically builds on pre-trained language models, and applies contrastive learning with losses such as Triplet Loss [123], SimCSE Loss [38], and Angle Loss [82]. These methods require labeled positive and negative pairs, which are absent in general-purpose corpora like C4 [120], leading researchers to rely on task-specific datasets such as Semantic Textual Similarity (STS), Natural Language Inference (NLI), and Information Retrieval (IR), or multi-task combinations.

**Semantic Textual Similarity (STS)** Datasets like STS17 [19] offer fine-grained similarity signals and have been widely used in models like Sentence-BERT [123] and TSDAE [153]. However, their small scale and narrow domain coverage often lead to overfitting and poor generalization [102].

**Natural Language Inference (NLI)** Datasets such as SNLI [16], MultiNLI [162], ANLI [104], and QA-NLI [28] annotate sentence pairs with *entailment*, *contradiction*, or *neutral*. These are widely used in models like Sentence-BERT [123], TSDAE [153], E5 [154], UDEVER [177], Angle [82], and GritLM [177]. While these datasets offer greater scale and domain diversity, the semantic signals often reflect shallow equivalence. For instance, SNLI pairs like "*A boy is jumping on a skateboard*" and "*The boy does a skateboarding trick*" fail to probe deeper pragmatic intent [16].

**Information Retrieval (IR)** Datasets, notably MS MARCO [9] and Natural Questions (NQ) [70], dominate retrieval-based training. Models like GTR [103], E5 [154], UDEVER [177], GritLM [101], and NV-Retriever [99] incorporate these datasets. The multilingual model mGTE [178] further draws

from sources including HotpotQA [170] and MIRACL [180]. Though useful for modeling lexical relevance, they reward literal matching and overlook implicit cues like stance or ideology.

**Multi-Task Learning** To improve generalization, models like mGTE [178] and the Jina Embeddings series [42, 43, 137] leverage multi-task corpora like C-MTP [166] and MEDI2 [101], which integrate STS, NLI, IR, QA, and other relevant data in pair or triplet format. While these datasets broaden coverage across tasks and domains, they still largely omit examples involving pragmatic inference, speaker stance, or sociocultural context, which are the key elements of implicit meaning.

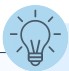 **Takeaway: Training Data Emphasize Surface Semantics**

Despite innovations in architecture and supervision, current training pipelines remain anchored in datasets that prioritize surface-level similarity. While multi-task learning expands coverage, implicit semantics, such as pragmatics, stance, and social context, remain largely absent from the training signal.

# 5 Benchmarks Do Not Evaluate Implicit Semantics

Despite the growth of large-scale benchmark suites ranging from semantic similarity and retrieval to multi-task generalization, most evaluations remain focused on surface-level semantics. This section surveys widely used benchmarks, including STS datasets, retrieval-centric benchmarks like BEIR, comprehensive multi-task suites such as MTEB, and emerging Retrieval-Augmented Generation (RAG) evaluations. While these resources provide broad coverage across tasks, domains, and languages, they rarely assess how well models capture implicit, contextual, or socially situated meaning, leaving a critical gap in current evaluation practices.

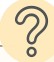 **Research Question: Evaluation Gap**

How are current text embedding models evaluated, and why do existing benchmarks fall short in capturing implicit meaning?

**Semantic Textual Similarity Benchmarks** STS tasks measure alignment between model-predicted similarities and human-annotated semantic similarity scores, using metrics like Spearman correlation. Popular datasets include STS12–17 [4, 5, 2, 1, 3, 19], STS-B [94], and SICK-R [93], all included in the MTEB benchmark [102]. Related binary classification tasks include MRPC [31], QQP [130], and GIS [82]. Although STS tasks are theoretically capable of evaluating deeper meaning, most focus on lexical variation and syntactic paraphrasing. Their scope is limited by construction methods and annotator biases, and they were developed largely before the rise of LLMs. As a result, they fail to probe pragmatic, attitudinal, or culturally embedded semantics.

**Information Retrieval Benchmarks** IR benchmarks assess how well models retrieve relevant documents using embedding similarity. Performance is measured using ranking metrics such as MRR, nDCG@$k$, and Recall@$k$ [160, 148]. Datasets like MS MARCO [9], NQ [70], SQuAD [121], Mr. TyDi [179], and MIRACL [180] are commonly used, with BEIR [148] aggregating 18 such datasets across diverse retrieval scenarios. Newer domain- and language-specific benchmarks include BRIGHT [139], LitSearch [6], COIR [81], the Scandinavian Benchmark [35], BEIR-NL [11], and RusBEIR [68]. Despite impressive coverage across domains and languages, IR tasks mostly evaluate surface-level relevance and do not test whether models capture deeper semantic alignment. Tasks like retrieving documents that match a speaker's stance or ideological framing remain underexplored.

**Multi-Task Benchmarks** MTEB [102] is a leading benchmark suite spanning 58 datasets and 8 task types. Variants such as C-MTEB [166], MMTEB [34], ArabicMTEB [14], PL-MTEB [118], and FaMTEB [186] expand coverage across languages, while domain-specific extensions like SciRepE-val [133], FinMTEB [147], ChemTEB [58], and MIEB [164] target specific verticals. Challenging tasks like reasoning and instruction-following have been introduced in ATEB [44]. Crowdsourced platforms like MTEB Arena[1] and Search Arena[2] provide user-driven comparisons across tasks [129].

---

[1] https://huggingface.co/spaces/mteb/arena
[2] https://blog.lmarena.ai/blog/2025/search-arena/

Despite offering flexible, model-agnostic evaluation, these platforms still rely on traditional metrics and rarely test for implicit meaning. In practice, only a few MTEB datasets go beyond surface semantics, limiting their value for evaluating interpretive depth.

**Retrieval-Augmented Generation (RAG) Benchmarks** RAG benchmarks evaluate how well embeddings retrieve relevant content to support generative tasks. Benchmarks such as RGB [24], CRUD-RAG [90], DomainRAG [158], MultiHop-RAG [146], , LegalBench-RAG [116], MIRAGE [168] and CyberMetric [151], cover multilingual, domain-specific, and multi-hop scenarios. Other general-purpose tools include RAGBench [36], RAGTruth [105], UDA [51], and toolkits like BERGEN [122]. Recent proposals like eRAG [124] and another MIRAGE [111] support fine-grained retrieval evaluation. While RAG setups touch on reasoning and hallucination, they still prioritize factual retrieval. As a result, the underlying semantic evaluations resemble IR tasks, offering limited insight into how well embeddings reflect implicit intent, stance, or social meaning.

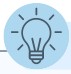

**Takeaway: Current Evaluation Focuses Mostly on Surface Semantics**

Current benchmarks provide broad task and domain coverage, but overwhelmingly emphasize surface-level similarity and relevance. They rarely assess a model's ability to capture implicit meaning, such as pragmatics, stance, or social context, leaving a critical gap in how we evaluate semantic understanding.

# 6 Empirical Evidences

To provide empirical evidence and motivate future research, we conduct a pilot study evaluating whether state-of-the-art embedding models can effectively capture implicit semantics.

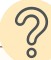

**Research Question: Empirical Gap**

Do state-of-the-art embedding models effectively capture implicit meaning across utterance, speaker, and society levels?

**Experimental Setup** We evaluate embeddings on seven datasets spanning three tiers of implicit meaning: (1) Utterance level: Pragmatics Understanding Benchmark (PUB), including Implicature (**P-IMP**), Presupposition (**P-PRE**), and Reference & Deixis (**P-R&D**) [136, 89, 182, 87, 21, 53, 112]; (2) Speaker level: **P-Stance** dataset [84] for stance detection; and (3) Society level: the datasets of Implicit Hate Speech (**IHS**) [33], Social Bias Inference Corpus (**SBIC**) [126], and Political Bias (**Pol. Bias**) [10]. Together, these datasets provide a structured view of implicit meaning.

Since these datasets were not originally designed for embedding evaluation, we reformulate them into classification, pairwise classification (following the MTEB benchmark [102]), and zero-shot formats, where models select the label with the highest embedding similarity. We test models from four representative categories: encoder-only models, LLM-based models, multimodal encoder models, and proprietary embeddings (OpenAI). Bag-of-Tokens [45, 141] and random baselines are included for comparison. Implementation details are provided in Appendix A.1.

**Results and Analysis** As depicted in Table 1, encoder-only models often perform only marginally better than Bag-of-Tokens and random baselines. LLM-based models and OpenAI embeddings generally achieve stronger results. Although OpenAI models rank lower on MTEB, they perform well on these implicit semantics datasets, highlighting a potential disconnect between benchmark performance and deeper semantic competence.

Moreover, performance varies by semantic tier. As shown, **Linq-Mistral** excels in utterance-level tasks, **OpenAI-Large** leads in speaker and societal datasets, and **E5-Mistral** shows strength in political bias detection. These differences suggest that current models may specialize in different semantic dimensions, revealing fragmentation in their implicit meaning capabilities.

Overall, these observations affirm this paper's central claim: *state-of-the-art embedding models remain limited in capturing implicit semantics.* High MTEB scores do not translate to robustness on tasks involving pragmatic inference, stance, or social context. The fact that many models barely surpass Bag-of-Tokens underscores a fundamental evaluation gap.

| Model | Utterance Level | | | Speaker Level | Society Level | | | Avg. Acc. ↑ |
|---|---|---|---|---|---|---|---|---|
| | P-IMP | P-PRE | P-R&D | P-Stance | IHS | SBIC | Pol. Bias | |
| **Random** | 48.5 | 54.1 | 38.8 | 51.3 | 27.5 | 59.2 | 34.5 | 44.8 |
| **Bag-of-Tokens** [141] | 56.5 | 75.3 | 48.2 | 73.4 | 59.6 | 80.7 | 41.6 | 62.2 |
| *Encoder Only Models* | | | | | | | | |
| **S-BERT** [123] | 61.7 | 72.8 | 55.7 | 72.9 | 60.8 | 81.8 | 47.9 | 64.8 |
| **GIST-Small** [134] | 65.8 | 76.1 | 58.8 | 76.0 | 61.8 | 81.6 | 49.1 | 67.0 |
| **BGE-Base** [167] | 65.0 | 75.6 | 57.3 | 74.4 | 62.9 | 82.1 | 52.1 | 67.1 |
| **Angle** [82] | 69.4 | 78.8 | 57.2 | 76.4 | 59.5 | 83.7 | 50.4 | 67.9 |
| **BGE-Large** [167] | 68.3 | 75.5 | 58.1 | 76.0 | 63.5 | 83.4 | 51.5 | 68.0 |
| **MXBAI-Large** [74] | 69.8 | 78.2 | 59.4 | 75.6 | 60.1 | 83.6 | 50.5 | 68.2 |
| **GIST-Large** [134] | 68.8 | 76.6 | 62.9 | 76.7 | 64.2 | 83.4 | 52.5 | 69.3 |
| **Stella** [175] | 72.1 | 81.5 | 59.6 | 76.5 | 60.4 | 84.0 | 54.4 | 69.8 |
| *Large Language Models* | | | | | | | | |
| **Linq-Mistral** [65] | **80.3** | **87.7** | **70.4** | 75.8 | 61.4 | 82.0 | 56.8 | 73.5 |
| **E5-Mistral** [155, 154] | 78.1 | 81.8 | 63.4 | 81.1 | 63.9 | 84.8 | **71.5** | 74.9 |
| **GTE-Qwen** [85] | 73.4 | 87.3 | 68.1 | 80.9 | 65.6 | 84.5 | 66.8 | 75.2 |
| *Multimodal Encoder Models* | | | | | | | | |
| **Jasper** [175] | 73.3 | 80.1 | 63.0 | 80.1 | 65.7 | 84.2 | 63.9 | 72.9 |
| *Proprietary Models* | | | | | | | | |
| **OpenAI-Small** | 71.3 | 78.1 | 64.3 | 80.0 | 66.2 | 83.9 | 56.6 | 71.5 |
| **OpenAI-Large** | 76.0 | 80.2 | 66.4 | **83.7** | **67.1** | **85.4** | 66.3 | 75.0 |

Table 1: Average accuracy (%) of embedding models across seven datasets representing three tiers of implicit semantics: utterance level (pragmatics), speaker level (stance), and societal level (social meaning). Results highlight differences in model capabilities across semantic levels and underscore the challenges of capturing implicit meaning.

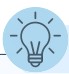
**Takeaway: Empirical Evidence Highlights the Implicit Semantics Gap**

Despite strong performance on standard benchmarks, embedding models struggle with tasks involving implicature, stance, and social meaning. Their inconsistent performance across semantic tiers and the proximity to Bag-of-Tokens baselines underscores the need for new training and evaluation strategies that directly target implicit semantics.

# 7 Towards Embeddings that Capture Implicit Meaning

To address the implicit semantics gap, we propose three complementary directions: enriching training data, designing targeted benchmarks, and treating implicit meaning as a core modeling objective. Together, these steps can guide the development of embeddings that go beyond surface-level similarity.

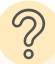
**Research Question: Research Agenda**

What steps should be taken to enable text embeddings to capture implicit meaning?

## 7.1 Curating More Diverse Training Data

Training data fundamentally shape what embedding models learn. As the adage goes, "*garbage in, garbage out*"–surface-level inputs yield surface-level representations. To enable models to capture implicit meaning, we must expand beyond narrow datasets and embrace greater linguistic, cultural, and contextual diversity. Beyond manual curation, recent advances in LLM-based data generation offer promising directions. Prior work has used LLMs to synthesize training examples for embedding models [155, 22]; future efforts should guide this generation toward phenomena like implicature, presupposition, and stance.

Linguistic theory provides a rich foundation for this endeavor. Decades of research have outlined typologies of implicit meaning, which can inform the design of more semantically grounded training

signals. Aligning synthetic data with these frameworks can help embeddings internalize meanings rooted in pragmatics and social context–dimensions often absent from existing datasets.

### 7.2 Designing Benchmarks for Implicit Meaning

Benchmarks drive progress by defining what models are expected to learn. However, existing suites like MTEB primarily test surface similarity. Their open-source nature has also led to data leakage and leaderboard inflation, weakening their value as generalization tests. This shift toward leaderboard optimization deviates from the original goal of embeddings: producing general-purpose, transferable representations. Among MTEB's 58 tasks, only a few probe beyond surface meaning, and even recent additions like ATEB emphasize reasoning or safety over pragmatic and cultural nuance.

New benchmarks should be explicitly constructed to test underrepresented forms of meaning. Tasks should include inference from indirect cues, stance recognition, and sociolinguistic variation, reflecting the interpretive demands of real-world language understanding.

### 7.3 Framing Implicit Semantics as a Modeling Goal

A deeper challenge is that implicit meaning is rarely treated as a first-class modeling objective. While LLM research increasingly investigates contextual, attitudinal, and social understanding [79, 78, 59, 136, 173, 27, 142, 91], embedding models remain optimized for benchmarks that reward superficial similarity. This misalignment leads models to optimize for what is easy to measure over what is meaningful to understand. Without explicitly targeting implicit semantics, advances in architecture and supervision risk reinforcing shallow representations. Reframing modeling goals around deeper semantic dimensions can produce embeddings that more faithfully reflect human communication.

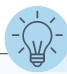

> **Takeaway: Future Opportunities for Text Embedding Research**
>
> To move forward, text embedding research must embrace implicit meaning as a central objective. This includes: (1) curating linguistically informed and culturally diverse training data, (2) designing benchmarks that evaluate pragmatic, attitudinal, and social understanding, and (3) reframing implicit semantics as a core modeling goal. Such a shift will lead to more robust, context-aware representations for real-world applications.

## 8 Alternative Views

While this paper advocates for embedding models to capture implicit semantics, alternative perspectives support maintaining the current focus on surface-level similarity. One argument is that for many practical tasks, such as search, recommendation, or clustering, surface semantics are often sufficient. Incorporating deeper meaning may add complexity without clear benefits.

Another view holds that pragmatic and socially grounded meaning is better handled by LLMs, which are explicitly designed for contextual reasoning and discourse-level understanding. In contrast, embeddings are valued for their efficiency and general-purpose utility. From this perspective, expecting embeddings to model implicit meaning may blur their role and dilute their purpose.

## 9 Conclusions

Despite significant progress in text embedding research, current models remain narrowly focused on surface-level semantics, failing to capture the implicit meanings that are central to human communication. This paper calls for a paradigm shift: embedding models must move beyond lexical similarity to explicitly model pragmatic, attitudinal, and sociocultural meaning. Drawing from linguistic theory, we propose a three-tier framework for implicit meaning and present empirical evidence that state-of-the-art models struggle with tasks requiring deeper interpretive reasoning. To advance the field, we advocate for semantically richer and more diverse training data, benchmarks that directly evaluate implicit understanding, and a reframing of implicit semantics as a core modeling objective. Embeddings that capture these deeper dimensions will enable more robust, context-aware systems aligned with the complexity of real-world language.

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

## A  Experiment Details

### A.1  Implementation Details

**Checkpoints** Table 2 lists the model checkpoints used in the experiments presented in Section 6. We adopt the official checkpoints used by the MTEB benchmark and evaluate all models using the default settings from the Sentence Transformers library [123], without additional parameter tuning or prompting. For OpenAI's proprietary models, we obtain embeddings using OpenAI's official client library.[3] A random baseline is implemented by sampling predictions according to the label distribution of each dataset. For the baseline of Bag-of-Tokens [45, 141], we use the `google-bert/bert-base-uncased` tokenizer.

| Model | Model Size | Checkpoint |
|---|---|---|
| **S-BERT** [123] | 22.7M | `sentence-transformers\all-MiniLM-L6-v2` |
| **GIST-Small** [134] | 33.4M | `avsolatorio\GIST-small-Embedding-v0` |
| **BGE-Base** [167] | 109M | `BAAI\bge-base-en-v1.5` |
| **Angle** [82] | 335M | `WhereIsAI\UAE-Large-V1` |
| **BGE-Large** [167] | 335M | `BAAI\bge-large-en-v1.5` |
| **MXBAI-Large** [74] | 335M | `mixedbread-ai\mxbai-embed-large-v1` |
| **GIST-Large** [134] | 335M | `avsolatorio\GIST-large-Embedding-v0` |
| **Stella** [175] | 435M | `NovaSearch\stella_en_400M_v5` |
| **Linq-Mistral** [65] | 7.11B | `Linq-AI-Research\Linq-Embed-Mistral` |
| **E5-Mistral** [155, 154] | 7.11B | `intfloat\e5-mistral-7b-instruct` |
| **GTE-Qwen** [85] | 7.61B | `Alibaba-NLP\gte-Qwen2-7B-instruct` |
| **Jasper** [175] | 1.99B | `NovaSearch\jasper_en_vision_language_v1` |
| **OpenAI-Small** | Unknown | `text-embedding-3-small` |
| **OpenAI-Large** | Unknown | `text-embedding-3-large` |

Table 2: List of models and their corresponding checkpoints.

**Tasks** We evaluate a diverse set of tasks designed to capture different aspects of implicit semantics. Due to data format differences, we organize them into three evaluation settings: **classification**, **pair classification**, and **zero-shot classification**. Each setting includes the following datasets:

- **Classification:** From the **Pragmatics Understanding Benchmark (PUB)**, we include *Task 1 (Direct/Indirect Classification)*, *Task 2 (Response Classification without Implied Meaning)*, *Task 3 (with Implied Meaning)*, *Task 6 (Understanding Sarcasm)*, *Task 10 (Implicature NLI)*, *Task 11 (Presupposition NLI)*, *Task 12 (Presupposition over QA)*, and *Task 13 (Deictic QA)*. We also evaluate all three subsets of the **P-Stance** dataset–*Trump*, *Biden*, and *Bernie*–for stance classification. For the **Implicit Hate Speech (IHS)** dataset, we include *detection*, *categorization*, and *target identification* tasks. For the **Social Bias Inference Corpus (SBIC)**, we evaluate five binary classification tasks: *whoTarget* (whether the target is a group), *intentYN* (intent to offend), *sexYN* (presence of sexual content), *offensiveYN* (offensiveness), and *hasBiasedImplication* (biased implications). Lastly, we include the **Political Bias (Pol. Bias)** classification dataset.
- **Pair Classification:** We adapt *Task 5 (Agreement Detection)* from **PUB**.
- **Zero-shot Classification:** We include *Task 4 (Implicature Recovery)*, *Task 7 (Figurative Language Understanding — No Hint)*, *Task 8 (with Positive Hint)*, *Task 9 (with Contrastive Hint)*, and *Task 14 (Reference via Metonymy)* from **PUB**.

**Evaluation Protocols** For classification and pair classification tasks, we follow the standard protocol from the MTEB benchmark [102]. For zero-shot classification, we adopt the embedding-based approach described in OpenAI's documentation,[4] where both the input question and text are embedded together, and each answer option is embedded separately. The answer option with the highest similarity to the input is selected as the prediction.

---

[3]`https://platform.openai.com/docs/api-reference/embeddings`
[4]`https://platform.openai.com/docs/guides/embeddings#use-cases`

| Model | PUB | | | | | | | | | | | | | |
|---|---|---|---|---|---|---|---|---|---|---|---|---|---|---|
| | Implicature | | | | | | | | | | Presupposition | | Ref. & Deixis | |
| | T1 | T2 | T3 | T4 | T5 | T6 | T7 | T8 | T9 | T10 | T11 | T12 | T13 | T14 |
| **Random** | 48.0 | 49.2 | 52.9 | 24.9 | 50.0 | 51.0 | 49.2 | 47.1 | 49.7 | 62.9 | 36.4 | 71.7 | 56.0 | 21.7 |
| **Bag-of-Tokens** [141] | 81.2 | 69.4 | 78.2 | 29.3 | 51.0 | 12.5 | 50.4 | 74.7 | 32.7 | 86.0 | 66.9 | 83.6 | 64.5 | 31.9 |
| | *Encoder Only Models* | | | | | | | | | | | | | |
| **S-BERT** [123] | 82.0 | 72.4 | 83.1 | 35.4 | 53.5 | 34.0 | 59.9 | 78.6 | 39.0 | 79.0 | 60.0 | **85.6** | 68.5 | 42.9 |
| **GIST-Small** [134] | 74.8 | 74.2 | 83.3 | 44.5 | 56.1 | 40.2 | 67.4 | 86.8 | 52.4 | 77.9 | 66.9 | 85.2 | 68.5 | 49.0 |
| **BGE-Base** [167] | 83.0 | 71.5 | 79.6 | 33.2 | 56.0 | 42.8 | 69.8 | 80.9 | 55.6 | 77.6 | 66.1 | 85.2 | 68.0 | 46.6 |
| **Angle** [82] | 97.2 | 77.3 | 81.2 | 41.5 | 58.1 | 32.8 | 75.1 | 82.8 | 60.7 | 87.6 | 74.2 | 83.4 | 66.0 | 48.4 |
| **BGE-Large** [167] | 87.0 | 76.8 | 79.8 | 43.0 | 57.7 | 41.8 | 75.3 | 84.0 | 59.6 | 77.9 | 65.8 | 85.2 | 68.0 | 48.1 |
| **MXBAI-Large** [74] | 97.2 | 78.4 | 81.0 | 41.2 | 58.5 | 34.0 | 75.5 | 83.3 | 61.2 | 87.4 | 73.6 | 82.8 | 67.5 | 51.2 |
| **GIST-Large** [134] | 79.6 | 76.8 | 83.1 | 46.9 | 57.9 | 37.8 | 78.0 | 88.5 | 61.5 | 78.3 | 68.1 | 85.2 | 71.5 | 54.3 |
| **Stella** [175] | 95.8 | 79.8 | 81.9 | 51.1 | 60.0 | 35.5 | 76.2 | 87.7 | 60.8 | 91.7 | 83.6 | 79.3 | 66.0 | 53.2 |
| | *Large Language Models* | | | | | | | | | | | | | |
| **Linq-Mistral** [65] | 99.6 | **89.3** | **88.6** | 47.5 | **70.0** | 67.0 | **88.6** | **94.5** | 61.4 | **96.2** | **91.4** | 84.0 | 74.0 | **66.7** |
| **E5-Mistral** [155, 154] | 97.2 | 87.2 | 85.8 | 43.8 | 69.5 | **69.0** | 87.7 | 92.7 | **62.9** | 85.0 | 78.3 | 85.2 | 68.0 | 58.9 |
| **GTE-Qwen** [85] | **100.0** | 87.7 | 87.5 | 43.6 | 61.8 | 63.0 | 69.0 | 82.7 | 48.5 | 89.8 | 89.4 | 85.2 | **78.0** | 58.2 |
| | *Multimodal Encoder Models* | | | | | | | | | | | | | |
| **Jasper** [175] | 97.8 | 84.2 | 85.4 | 50.6 | 61.6 | 49.8 | 78.0 | 88.4 | 55.4 | 81.9 | 75.0 | 85.2 | 70.5 | 55.6 |
| | *Proprietary Models* | | | | | | | | | | | | | |
| **OpenAI-Small** | 98.8 | 79.4 | 84.9 | **56.0** | 56.7 | 35.5 | 79.3 | 89.9 | 55.0 | 78.1 | 71.1 | 85.2 | 73.0 | 55.6 |
| **OpenAI-Large** | 99.6 | 87.7 | 87.7 | 50.8 | 61.9 | 55.5 | 83.9 | 91.8 | 58.4 | 83.1 | 75.3 | 85.2 | 73.5 | 59.3 |

Table 3: The accuracy (%) of embedding models on the Pragmatics Understanding Benchmark (PUB) tasks. Each task is labeled as T1–T14, corresponding to the 14 tasks in the PUB benchmark.

## A.2 Additional Results

The complete results, including the accuracy (%) for individual task, are presented in Tables 3 and 4. The values reported in Table 1 are computed by averaging across tasks within each dataset.

**Widespread Variance Across Models** The results reveal inconsistent performance across embedding models. For example, many models achieve near-perfect accuracy on *Task 1 (Direct/Indirect Classification)* from PUB, while models such as **GIST-Small**, **S-BERT**, and **BGE-Base** perform only marginally better or even worse than the Bag-of-Tokens baseline. Similarly, on *Task 10 (Implicature NLI)*, several models, including OpenAI's proprietary models and the LLM-based **E5-Mistral** and **Jasper**, underperform the Bag-of-Tokens baseline. These findings demonstrate that strong performance on surface-level benchmarks does not reliably transfer to tasks requiring deeper semantic understanding.

**Strengths of Large and Multimodal Models** Large and multimodal models tend to lead in overall performance. **Jasper**, for example, ranks among the top across a wide range of tasks, particularly within **IHS** and **SBIC**. Similarly, large-scale models such as **E5-Mistral** and **OpenAI-Large** perform well across domains, excelling in social bias classification and pragmatics reasoning. These results suggest that increased model size contributes positively to handling complex semantic phenomena.

**Persistent Challenges in Implicature and Reference Tasks** Despite their strengths, even the largest models struggle with specific pragmatic tasks. Notably, **Task 4 (Implicature Recovery)** remains difficult across all models, with scores rarely exceeding 50%. Even top-tier models like **GTE-Qwen** and **OpenAI-Large** achieve only modest gains over Bag-of-Tokens. These findings point to a fundamental limitation in how current training pipelines address implicit meaning.

**Implications for Benchmark and Model Design** In summary, these results reveal persistent blind spots in current embedding models, particularly for tasks involving implicature, figurative language, presupposition, and social inference. Addressing these challenges will require more linguistically grounded training strategies and benchmark datasets that explicitly target underexplored aspects of implicit meaning.

| Model | P-Stance | | | IHS | | | SBIC | | | | | Pol. Bias |
|---|---|---|---|---|---|---|---|---|---|---|---|---|
| | Trump | Biden | Bernie | Det. | Cat. | Tar. | Tar. | Intent | Sex | Off. | Bias | |
| **Random** | 51.7 | 50.9 | 51.2 | 52.5 | 16.6 | 13.4 | 48.7 | 58.2 | 76.6 | 62.0 | 50.5 | 34.5 |
| **Bag-of-Tokens** [141] | 74.6 | 75.4 | 70.1 | 74.5 | 55.0 | 49.3 | 77.7 | 76.1 | 92.0 | 79.6 | 78.1 | 41.6 |
| *Encoder Only Models* | | | | | | | | | | | | |
| **S-BERT** [123] | 72.1 | 77.4 | 69.3 | 73.2 | 58.0 | 51.2 | 78.8 | 78.0 | 91.9 | 81.2 | 79.1 | 47.9 |
| **GIST-Small** [134] | 76.6 | 78.4 | 72.9 | 74.0 | 59.5 | 51.9 | 77.9 | 77.7 | 93.1 | 81.3 | 78.0 | 49.1 |
| **BGE-Base** [167] | 74.5 | 78.8 | 69.9 | 74.2 | 60.7 | 53.9 | 78.7 | 78.5 | 92.6 | 82.0 | 78.6 | 52.1 |
| **Angle** [82] | 77.2 | 79.9 | 72.1 | 75.9 | 56.0 | 46.6 | 80.4 | 80.4 | 93.3 | 83.7 | 80.5 | 50.4 |
| **BGE-Large** [167] | 75.8 | 80.0 | 72.1 | 75.1 | 61.4 | 53.9 | 80.3 | 79.9 | 93.3 | 83.0 | 80.6 | 51.5 |
| **MXBAI-Large** [74] | 76.4 | 78.9 | 71.5 | 76.4 | 56.5 | 47.3 | 80.4 | 80.3 | 93.1 | 83.6 | 80.4 | 50.5 |
| **GIST-Large** [134] | 76.8 | 80.1 | 73.2 | 75.0 | 63.1 | 54.4 | 80.5 | 79.3 | 93.3 | 82.9 | 80.9 | 52.5 |
| **Stella** [175] | 79.2 | 80.0 | 70.2 | 76.9 | 56.7 | 47.6 | 81.2 | 80.9 | 93.4 | 83.2 | 81.5 | 54.4 |
| *Large Language Models* | | | | | | | | | | | | |
| **Linq-Mistral** [65] | 79.4 | 78.8 | 69.3 | 75.1 | 57.8 | 51.2 | 79.7 | 79.1 | 89.8 | 81.9 | 79.8 | 56.8 |
| **E5-Mistral** [155, 154] | 84.8 | 82.3 | 76.1 | 79.2 | 61.7 | 50.9 | 82.0 | **82.5** | 93.3 | 83.9 | 82.1 | **71.5** |
| **GTE-Qwen** [85] | 83.8 | 82.0 | 76.9 | 79.1 | 63.2 | 54.5 | 82.1 | 80.6 | 94.0 | 83.7 | 82.0 | 66.8 |
| *Multimodal Encoder Models* | | | | | | | | | | | | |
| **Jasper** [175] | 81.6 | 82.6 | 76.2 | 78.4 | 64.6 | 54.1 | 81.4 | 81.2 | 93.9 | 83.1 | 81.5 | 63.9 |
| *Proprietary Models* | | | | | | | | | | | | |
| **OpenAI-Small** | 82.4 | 81.1 | 76.7 | 78.5 | 64.7 | **55.2** | 80.7 | 81.1 | 93.5 | 83.3 | 80.8 | 56.6 |
| **OpenAI-Large** | **87.5** | **83.8** | **79.7** | **80.2** | **67.3** | 53.7 | **82.9** | 82.3 | **94.2** | **84.7** | **83.1** | 66.3 |

Table 4: The accuracy (%) of embedding models on additional implicit meaning benchmarks. **P-Stance** includes stance detection tasks for Trump, Biden, and Bernie. **Implicit Hate Speech (IHS)** comprises detection (Det.), categorization (Cat.), and target identification (Tar.) tasks. The **Social Bias Inference Corpus (SBIC)** includes target (Tar.), intent (Int.), sexism (Sex.), offensiveness (Off.), and bias detection tasks. **Political Bias (Pol. Bias)** refers to the political ideology classification task.

