# OpenReview forum: "Text Embeddings Should Capture Implicit Semantics, Not Just Surface Meaning"
_NeurIPS.cc/2025/Position_Paper_Track — Submitted to NeurIPS 2025 Position Paper Track_

### Official Review · Reviewer_UhFX · 2025-07-31

**Significance:** 2
**Presentation:** 4
**Rating:** 4
**Confidence:** 5

**Summary:**

The authors' position aims to inform the audience about today's textual embedding models, which lack knowledge of implicit semantics. Indeed, all the current models have been trained on tasks, datasets, and with training procedures not designed to handle such information. Hence, the authors inspect the main definitions of implicit semantics on three different linguistic levels (i.e., Utterance, Speaker, and Society), survey the actual panorama of text-embedding models available, and design a novel benchmark to assess the evidence of existing gaps on those models, which includes datasets (still an ongoing work as a pilot study) and renewed evaluation approaches. The proposal highlights a decrease in performance on the experimentation of the current models, discussing shortcomings in the adopted training procedure, existing datasets, and evaluation frameworks. The authors claim to include implicit semantics directly as a model objective of text embeddings, despite enforcing training procedures and designing focused benchmarks to bridge the existing gap, deviating from the contemporary research direction.

**Strengths:**

- The structure of the paper is simple, concise, and effective.
- The authors' position is timely and exciting given the current literature.
- The reasoning and experimentation to support the position of the paper are well-designed.
- The topic is certainly relevant to the community (or at least to different sub-areas).
- The surveyed work in describing today's panorama of available text embedding models is comprehensive and highly relevant as a further research tool.
- The paper is well-written and easy to follow.

**Weaknesses:**

- The paper presents a lot of repetition regarding its central objective and suggested solutions throughout all the sections, which is space-consuming and weighs down the reading.
- Figure 1 is misleading or not comprehensively presented if compared with the results shown in Table 1.
- The proposal lacks meaningful examples to highlight the actual influence of implicit semantics (Section 2.1, 2.2).
- Final sections reflect some inconsistencies. In section 4, the authors state that training processes fail to capture implicit semantics, while they primarily discuss datasets and tasks, with training approaches only defined as titles for subsections. In section 5, the authors' focus remains on the task rather than inspecting benchmarks, except for the Semantic Textual Similarity (STS) task.
- Lacking information on the novel refined dataset and the engaged procedure to train or evaluate models (Appendix A.1 just lists a bunch of different tasks never explained to the reader, also by including a further level of implicit semantics analysis outside the three ones presented before).

Minor: Some inaccuracies in section 7 (please see Questions)

**Questions:**

1) Do the authors consider the concept of LLMs from text embedding as different (from what emerges in 7.3, line 307)?
2) Is the example in section 2.1 more related to the emotions conveyed rather than implicit semantics? (The unexpected event of passing the test is related to astonishment). Moreover, the sentence "Sam quit smoking" already presents in the surface semantics of the verb quit implies that something has reached an end. Is there a formal definition that further justifies this sample or a different one?
3) What emerges is that tasks like IR, RAG, and some multi-tasking were not designed to handle implicit semantics, which is quite normal since their interest lies in retrieving documents. Can the authors elaborate?
4) Why do the authors say that many models barely surpass Bag-of-Tokens, while the performances of most of the models, as depicted in Table 1, show a large margin of difference? Are these results showing that existing models, especially contextualized embedding models, can already manage implicit semantics?
5) How do the authors intend to improve the model's learning of implicit semantics by using synthetic data generated by models that have not yet learnt them?

**Alternative Position:**

No

**Author Identification:**

No.

**Context:**

4

**Discussion:**

3

**Ethics:**

["NO or VERY MINOR ethics concerns only"]

**Position:**

Yes, the paper argues for or against a position related to machine learning.

**Support:**

1

**Thoroughness:**

5

---

### Official Review · Reviewer_mb9D · 2025-08-08

**Significance:** 3
**Presentation:** 3
**Rating:** 7
**Confidence:** 4

**Summary:**

This is a position paper that argues the text embedding research community should shift its focus from surface-level semantics to implicit semantics as a core modeling objective. The authors state that while current text embedding models have made significant progress, they are primarily trained and evaluated on tasks that only capture shallow meaning, such as lexical overlap and syntactic variation. Drawing on linguistic theory, the paper proposes a three-tier framework for implicit meaning: utterance level (pragmatics), speaker level (stance), and society level (sociolinguistics). The paper includes a pilot study that demonstrates a performance gap, showing that state-of-the-art models perform only marginally better than a Bag-of-Tokens baseline on tasks that require implicit understanding. To address this gap, the authors call for a paradigm shift, advocating for the use of more diverse, linguistically grounded training data and the development of new benchmarks that explicitly evaluate deeper semantic understanding.

**Strengths:**

1. The paper has a clear and compelling argument that is well-supported by linguistic theory. It effectively highlights a critical, often overlooked, limitation in the current field of text embedding research.
2. The inclusion of a pilot study provides concrete evidence to back the central claim. The "performance gap" shown in Figure 1, where top models perform significantly worse on implicit semantics tasks compared to surface meaning tasks, is a powerful visual and empirical demonstration of the problem.
3. The three-tier framework (utterance, speaker, society) for implicit meaning is a strong organizational tool that helps the reader understand the different facets of the problem. It moves the discussion beyond a vague notion of "deeper meaning" into a specific, actionable analysis.
4. The paper provides a thorough review of existing text embedding models, training processes, and benchmark suites, effectively showing how these current practices reinforce the focus on surface semantics and neglect implicit meaning.

**Weaknesses:**

1. Limited Pilot Study: The empirical evidence is limited to a single "pilot study". While compelling, a more extensive experimental section with a wider range of models and datasets, and a more detailed description of the methodologies would strengthen the argument.
2. Lack of a Proposed Solution: The paper outlines what should be done, such as creating new benchmarks and using better training data, but it does not provide a concrete example of a new model architecture or training method that successfully captures implicit semantics. The proposed solutions are high-level recommendations rather than a detailed technical proposal.

**Questions:**

1. You mention the performance gap between current models and the Bag-of-Tokens baseline on implicit semantics tasks. Could you provide more detailed insights into the experimental setup and dataset choices used for your pilot study? A deeper understanding of your methodology might help clarify why the performance gap exists and whether it could be addressed through model adjustments
2. Given the focus on implicit semantics, how do you envision overcoming the challenge of ensuring that new, linguistically grounded datasets reflect the complexity of real-world language while also being scalable for large-scale training?
3. While your paper advocates for more diverse training data, what are some specific data augmentation strategies or techniques you foresee as being effective in training models to capture implicit meaning, such as pragmatics, stance, and sociocultural context?

**Alternative Position:**

Yes, and alternative positions are well-considered and addressed by the argument

**Author Identification:**

No.

**Context:**

3

**Discussion:**

3

**Ethics:**

["NO or VERY MINOR ethics concerns only"]

**Position:**

Yes, the paper argues for or against a position related to machine learning.

**Support:**

2

**Thoroughness:**

4

---

### Official Review · Reviewer_XgTJ · 2025-08-12

**Significance:** 2
**Presentation:** 3
**Rating:** 6
**Confidence:** 4

**Summary:**

This position paper argues that current text embedding models, despite strong performance on benchmarks like MTEB, primarily capture surface-level semantics such as lexical overlap, syntactic variation, and topical similarity, while failing to represent deeper implicit meaning shaped by pragmatics, speaker stance, and sociocultural context.
Drawing from linguistic theory, the authors propose a three-tier framework for implicit meaning: (1) utterance-level pragmatic inference, (2) speaker-level stance-taking, and (3) society-level sociolinguistic signals.
They present a pilot study using seven datasets across these tiers, showing that state-of-the-art embeddings perform only marginally better than a Bag-of-Tokens baseline, in contrast to their high scores on conventional benchmarks.
The paper advocates for curating more linguistically and culturally diverse training data, designing benchmarks that explicitly target implicit semantics, and reframing implicit meaning as a core modeling objective for embedding research.

**Strengths:**

* The paper addresses a timely and arguably underexplored gap in embedding research: the limited ability of current models to capture implicit meaning.
* The strong theoretical grounding in pragmatics, stance-taking, and sociolinguistics, clearly integrated into the motivation.
* It presents a well-structured problem framing that distinguishes surface-level semantics from deeper, context-dependent meaning.
* The empirical pilot study spans multiple semantic tiers (utterance, speaker, society) and model families (encoder-only, LLM-based, multimodal, proprietary).
* The authors proposes actionable high-level directions for the community, including richer training data, targeted benchmarks, and reframing modeling goals.

**Weaknesses:**

* From my perspective, the identified gap is largely application-specific. For many general-purpose LLM uses, the surface-level semantics captured by current embeddings are often sufficient, making the proposed shift less universally necessary.
* Empirical study is small-scale, using repurposed datasets rather than introducing a dedicated benchmark for implicit meaning.
* No concrete methodological innovations; contributions are primarily conceptual and programmatic.
* The paper does not fully address annotation cost, bias risks, or validation challenges for generating implicit meaning data, especially when using LLMs.

**Questions:**

* How would you design implicit semantics benchmarks to ensure broad coverage without becoming overly task-specific?
* Should implicit semantics be a universal goal for all embedding models, or should it be pursued only for application domains where it is clearly beneficial?
* Can you provide concrete downstream examples where improved implicit meaning capture in embeddings leads to measurable performance gains over current approaches?

**Alternative Position:**

No

**Author Identification:**

No.

**Context:**

3

**Discussion:**

3

**Ethics:**

["NO or VERY MINOR ethics concerns only"]

**Position:**

Yes, the paper argues for or against a position related to machine learning.

**Support:**

3

**Thoroughness:**

3

---

### Note · Authors · 2025-08-20

**1-11 Submit Again:**

Definitely yes

**1-1 Submission Process:**

4

**1-2 Next Year:**

For next year, two improvements would be especially valuable:

1. Aligned reviewing timeline: Synchronize the review process with the main track to ensure consistency and reduce scheduling pressure.

2. Discussion period: Introduce a stage for author–reviewer and reviewer–reviewer dialogue to enable clarification, constructive exchange, and stronger evaluations.

**1-3 Future Development:**

We recommend the same two improvements for the track's longer-term development:

1. Align the reviewing timeline with the main track.

2. Include a structured discussion period to encourage deeper engagement and more balanced feedback.

**1-4 Interest:**

["Panel discussions with other position paper authors", "Structured debates on controversial topics", "Workshops for developing position papers", "Mentorship programs for early-career researchers"]

**1-5 Thoughtful:**

7

**1-6 Supportive:**

8

**1-7 Technical Aspects Versus Position:**

4

**1-8 Gate Keeping:**

8

**1-9 Camera Ready Changes:**

If accepted, we will make the following changes based on the reviews:

1. Scope and Use Cases (§2): We will clarify our phenomenon-first, task-agnostic evaluation stance, explain why encoding implicit signals benefits IR/RAG, and emphasize that implicit semantics need not be a universal goal but broaden applicability in use cases. (Reviewers XgTJ and UhFX)

2. Experiments (§6): We will clarify the structure of the pilot study, expand experimental details, refine claims around Bag-of-Tokens, and clarify aggregation protocols. (Reviewers XgTJ, mb9D, and UhFX)

3. Future Directions (§7): We will elaborate the linguistically guided synthetic pipeline, expand the envisioned training objective that integrates three levels of implicit meaning, make the LLM–embedding distinction explicit, and extend the discussion of limitations. (Reviewers XgTJ, mb9D, and UhFX)

**3-1 Review Response1:**

XgTJ

**3-2 Reaction To Review1:**

We thank Reviewer XgTJ for the thoughtful and encouraging feedback. The review is supportive and inclusive, with a welcome focus on technical aspects. We appreciate the recognition of the paper's theoretical grounding in linguistics, the clarity of the three-tier framework, the value of the pilot, and the constructive questions that help make the contribution more actionable.

We will revise the manuscript along three lines:

1. Scope (§2): We will clarify that implicit semantics need not be a universal goal, but capturing such signals broadens applicability and supports concrete use cases such as diversified retrieval, safety, and stance-aware RAG.

2. Experiments (§6): We will foreground our phenomenon-first, task-agnostic assessment using generic similarity formats, paired with cross-domain and cross-register coverage to avoid domain lock-in.

3. Future Directions (§7): We will outline a linguistically guided data strategy with lightweight training recipes, and expand the discussion of limitations.

In summary, these revisions will sharpen the scope and utility of the paper, providing readers with a clear roadmap for evaluating implicit meaning and applying it where it delivers tangible benefits.

**3-3 Review Response2:**

mb9D

**3-4 Reaction To Review2:**

We thank Reviewer mb9D for the positive and constructive feedback. The review is supportive, thoughtful, and inclusive, with a welcome emphasis on technical aspects that make the position more actionable. We especially appreciate the clear articulation of strengths and the concrete questions regarding methodology, scalability, and training strategies.

We will revise the manuscript with the following three points:

1. Experiments (§6): We will clarify the structure of the pilot study, strengthen the main-text description, and connect it explicitly with Appendix A.1.

2. Data Construction (§7.1): We will elaborate on the linguistically guided synthetic pipeline, which generates contrastive minimal pairs by toggling triggers under hard constraints.

3. Training Objective (§7.3): We will expand our discussion of the envisioned objective, integrating the three levels of implicit meaning in linguistics.

In summary, these clarifications will provide readers with a clear and practical pathway from data preparation to model training for capturing implicit meaning.

**3-5 Review Response3:**

UhFX

**3-6 Reaction To Review3:**

We thank Reviewer UhFX for the rigorous and detail-oriented assessment. The review is supportive, thoughtful, and inclusive, with a welcome emphasis on technical aspects. It highlights clarifications and corrections that will sharpen the paper’s scope, evidence, and presentation.

We will revise the manuscript along three lines:

1. Use Cases (§2): We will add rationale for why encoding stance, hedged negation, and register benefits IR/RAG despite their traditional focus on topical relevance.

2. Experiments (§6): We will expand the setup and Appendix A.1 with concrete procedures, refine claims around Bag-of-Tokens, clarify aggregation protocols, and point to per-task results in Appendix A.2.

3. Future Directions (§7): We will elaborate the envisioned constrained synthetic data approach (§7.1) and make the LLM–embedding distinction explicit (§7.3).

In summary, these revisions aim to keep the manuscript compact while enhancing clarity, rigor, and practical guidance.

---

### Meta-Review · Area_Chair_pZ87 · 2025-09-06

**Rating:** 6
**Confidence:** 3

**Strengths:**

This paper addresses the shortcomings of current text embedding models that are trained mainly on surface level text and do not capture deeper semantics based on pragmatics, intent/stance and sociocultural context. The paper advocates having linguistically grounded training data along with better benchmarks that evaluated deeper semantics and not just surface level meaning.


Strengths:

1. The paper raises an important concern about current text embedding models and proposes an alternative in terms of culturally, pragmatically and linguistically grounded approach to training and evaluation.
2. The paper performs an interesting pilot study to highlight the shortcomings of current models with regard to pragmatic inference, stance detection and socio-cultural understanding.
3. The paper is well written, provides a solid position on the topic and provides actionable directions for researchers.

**Weaknesses:**

1. The empirical pilot study is done at a small scale by using existing datasets and no new benchmark is created.
2. It is not clear if the proposed deeper embedding models should be completely replace the existing surface level embedding models in all settings because it is not the case that existing embedding models are completely useless.
3.  A more in-depth empirical study is required to assess how much performance gain is achieved via deeper semantic embedding models.
4. The paper has writing inconsistencies in some parts, e.g., section 4, and section 7. These should be addressed based on reviewer comments.

**Questions:**

1. Authors should scale up the the pilot studies and empirically establish the claims.
2. Authors should resolve the inconsistencies in paper writing.
3. Address the questions raised by reviewers.

**Ethics:**

No ethical concerns raised.

**Thoroughness:**

1

---

### Decision · Program_Chairs · 2025-09-26

Reject